# Transition experiences of patients with post stroke dysphagia and family caregivers: A longitudinal, qualitative study

**Jian CHEN**[1,2☯], **Jianhui CHEN**[3☯], **Yuan WANG**[2], **Yanli CUI**[1,2], **Lin LIAO**[1,2], **Mingyu YAN**[1,2], **Yansi LUO**[1‡*], **Xiaomei Zhang**[1‡*]

1 Department of Neurology, Nanfang Hospital, Southern Medical University, Guangzhou City, Guangdong Province, China, 2 School of Nursing, Southern Medical University, Guangzhou City, Guangdong Province, China, 3 Department of Rheumatology, Nanfang Hospital, Southern Medical University, Guangzhou City, Guangdong Province, China

☯ These authors contributed equally to this work.
‡ YL and XZ also contributed equally to this work.
* a.yan24@163.com (YL); 2925611568@qq.com (XZ)

## Abstract

### Background

Stroke patients with dysphagia and family caregivers will experience multiple transitions during the whole process of the disease and various nursing needs will be generated. There is a lack of knowledge about their experiences at different transition stages. Thus, we aimed to explore the transition experiences of patients with post stroke dysphagia and family caregivers from admission to discharge home.

### Methods

A semi-structured interview based on Meleis's transition theory was used during hospitalization and telephone follow-up interviews were conducted in the first, third, and sixth month after the diagnosis of dysphagia. Interview transcripts were analyzed using the conventional content analysis method.

### Results

A total of 17 participants enrolled in the first face-to-face interview, 16 participants took part in the first month's telephone follow-up interview, 14 participants in the third month, and 12 participants in the sixth month. The transition experiences of patients with post stroke dysphagia and family caregivers could be summarized into three themes: (1)transition from onset to admission; (2)transition from discharge to other rehabilitation institutions; and (3) transition from discharge to home. Each theme had identified interrelated subthemes.

### Conclusions

The experiences of patients with post stroke dysphagia and family caregivers during transition are a dynamic process with enormous challenges in each phase. Collaboration with

**Data Availability Statement:** The data underlying the results of this study are available upon request due to ethical and legal restrictions under the medical ethics committee of NanFang hospital.

Interested researchers may contact the authors at 1329319886@qq.com or manager of medical ethics committee of NanFang hospital at nfyyec@163.com.

**Funding:** This work was supported by grants from the Natural Science Foundation of Guangdong Province (grant number 2022A1515012184).

**Competing interests:** The authors have declared that no competing interests exist.

health care professionals, follow-up support after discharge, and available community and social support should be integrated into transitional nursing to help patients facilitate their transition.

## Introduction

Dysphagia is a common clinical symptom in the elderly and neurological diseases. Statistics showed that the prevalence of dysphagia in the hospital setting was 36.5% [1] and that of the elderly in the community was 25.1% [2], while among stroke patients, the prevalence was as high as 42% [3]. Post stroke dysphagia (PSD) is a condition in which patients are unable to safely deliver food or liquid from the mouth to the stomach due to damaged motor neurons and reduced control of the oral and throat muscles caused by stroke. PSD not only affects the eating experiences of patients [4], but also causes weight loss and malnutrition [5], significantly reducing patients' quality of life. More seriously, if the food or liquid enters the respiratory tract, possible complications such as aspiration pneumonia and recurrent fever will have a direct impact on the patient's recovery outcome, and may even result in death [6]. Rehabilitation of PSD may last for a long period of time. Studies have shown that 11%~50% of patients with PSD still have no significant improvement in swallowing function after 6 months of neurological damage, which brings great challenges to patients and their families caregivers.

At present, the management of post stroke dysphagia mainly includes risk factors for dysphagia, early identification, management of aspiration, and telephone follow-up in most local healthcare settings [7, 8]. Studies on the experiences of PSD patients and caregivers mainly focuses on acute care during hospitalization and discharge to home. A qualitative study showed that PSD has many negative effects on patients' lives at home, and that they need long-term support from healthcare professionals, including ways to cope with the complications of PSD as well as psychological adjustment strategies for patients [9]. In addition, there is an unmet need for oral feeding skills, rehabilitation training methods, and assessment of nutritional status [10]. And the level of patients' care needs is closely related to mortality and prognosis [11]. Consequently, exploring patients' rehabilitation care needs plays a key role in facilitating their recovery. However, there is currently little research on the transition experiences of PSD patients and caregivers at various stages of recovery, as well as a lack of understanding of patients' care needs during the transition phases.

Transition is the process of changing from one stage, state or environment of life to another, referring to the outcome of complex human-environmental interactions, and also includes the psychological processes that occur to adapt to change [12]. The transition theory proposed by Meleis encompasses four major concepts: the nature of transitions; transition conditions; patterns of response; and nursing therapeutics [13]. Each major concept consists of several related subconcepts as well (see Fig 1). The theory reiterates the importance of comprehensive care and suggests that the transition will be accompanied by changes in life, health, interpersonal relationships, and environment, resulting in various nursing needs [14]. In addition, the theory provides an excellent perspective to systematically and comprehensively explore the nursing phenomena related to the transition process and gain an in-depth understanding of patients' transition experiences, to provide nursing strategies conducive to rehabilitation according to patients' individualized needs, possible difficulties, and concerns. Meleis's transition theory has been applied to different populations in nursing practice [15, 16]. Life changes for PSD patients and caregivers after the onset of stroke can also be seen as a series of

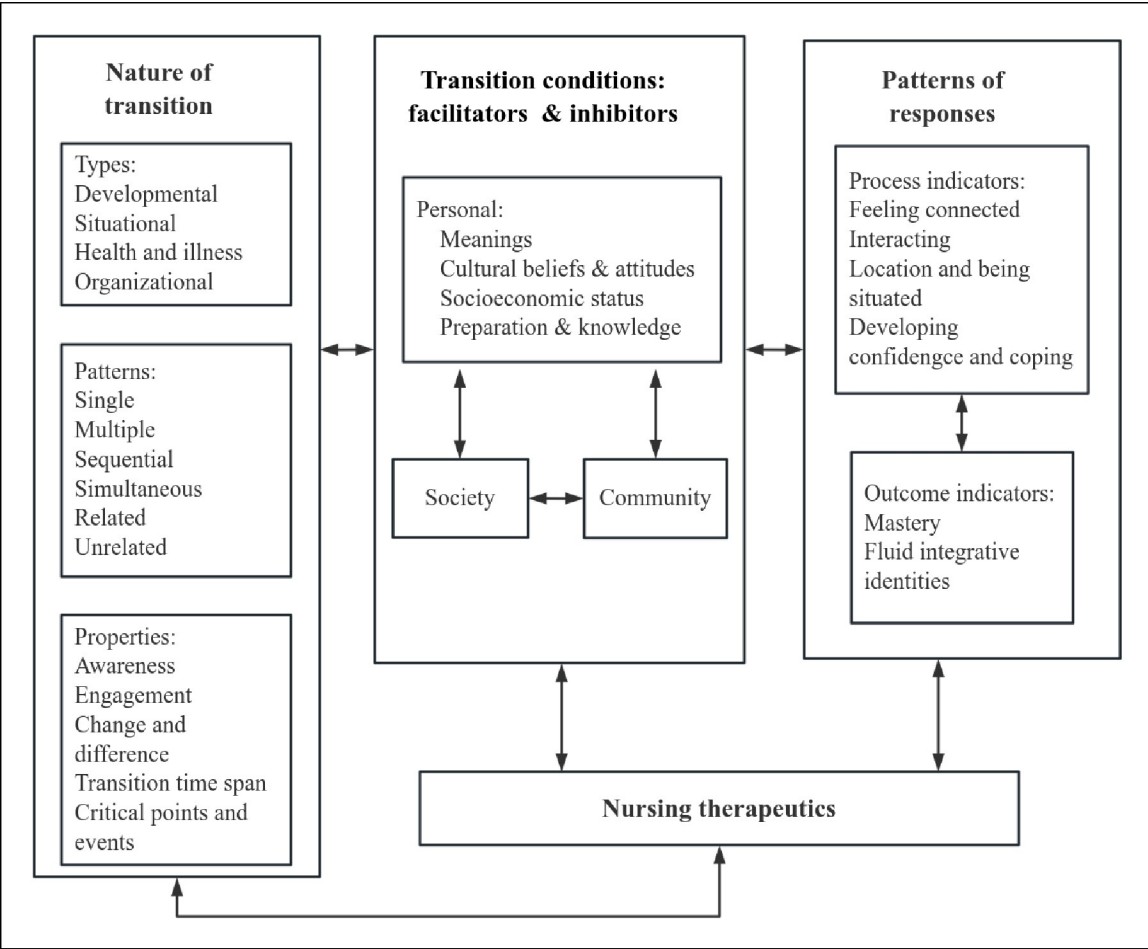

**Fig 1. The conceptual framework of Meleis's transition theory.**

transitions. So this study intends to use descriptive qualitative study to explore the experiences and nursing needs of PSD patients and family caregivers in different transition stages based on Meleis's transition theory, in order to help them achieve healthy transition and promote their rehabilitation.

## Methods

**Study design.** The study was designed and conducted based on the descriptive qualitative study, which is a research approach based in the philosophical tenets of naturalistic inquiry [17]. The features of descriptive qualitative study include recognition of varied shared experiences and the interactive-inseparable nature of human interaction and is analyzed using a content analysis method [18]. This approach helps to provide insight into the individual's experience and analyzed using conventional content analysis. We abided by the Consolidated Criteria for Reporting Qualitative Research(COREQ) [19] to ensure the explicit and comprehensive reporting of our findings, see S1 Checklist.

**Participants and settings.** The purposive sampling method was used to invite PSD patients and family caregiver who were admitted to the hospital from November to December, 2022 to participate in this study. A family caregiver was defined as someone close to the patient who was willing and able to live with the patient, for example a partner, family member or

siblings. We invited both patients and family caregivers to participate in the interviews. If the patient was able to communicate normally, the patient's interview results were mainly recorded, with the family caregiver supplementing, otherwise the family caregiver's interview results were recorded primarily. The first interview was face-to-face and took place in the hospital, while the follow-up interview after discharge was conducted by telephone and the patients might be transferred to other hospitals or discharged home.

Patients were eligible if they (1)met diagnostic criteria for ischemic stroke [20], confirmed by cranial CT or MRI; (2)dysphagia was evaluated by water swallow test combined with volume-viscosity swallow test-Chinese version [21]; (3)aged 18 years or older; (4)the vital signs were stable and communication was normal; (5)agreed to participate in the study. These inclusion criteria for the caregiver were: (1)undertook primary care of PSD patients and live with patients after discharge; (2)caregiving ≥4 hours per day; (3)agreed to participate in the study.

An exclusion criterion for patients were deafness, sensory aphasia, and pre-existing dysphagia due to other causes before a stroke. There were no exclusion criteria for family caregivers.

**Data collection.** We conducted a face-to-face interview with participants enrolled in this study, and then the transition experiences of participants after discharge were collected prospectively by telephone follow-up interviews. The first interview was a face-to-face interview with PSD patients and family caregivers during hospitalization to understand the transition experiences after admission. Two researchers were needed for this interview, one for the interview and one for the record. Interviews were audio-recorded with the consent of the participant, or transcribed throughout if consent was not given. Telephone follow-up was conducted in the first, third, and sixth month after the diagnosis of dysphagia in other rehabilitation hospitals or at home. Each interview lasted from 30 to 45 minutes. The duration of conversation was flexible and based on the participants' preference. Data were collected using a semi-structured interview consisting of open-ended questions based on Meleis's transition theory. The interview began with general questions, such as "Could you tell me something about your hospitalization experience? How do you feel?" and moved to more specific, detailed questions as the interview advanced, such as "When you were hospitalized, what was the most difficult thing for you? How was it solved?" The full topic guide is presented in S1 File.

The data collection process was terminated once the data reached saturation, which meant no additional information was likely to be gained [22].

**Ethics.** This study was one of the projects of Natural Science Foundation of Guangdong Province and was approved by the Medical Ethics Committee of NanFang Hospital of Southern Medical University before the commencement of the research. The approval number is NFEC-2022-403. In addition, after a detailed explanation of the purpose and methodology of the study, informed written consent of each participant for face-to-face interviews and follow-up was collected, and informed verbal consent was taken before starting interviews. Also, informed consent was obtained from their legal guardians in the case of involving the patient with unconsciousness. The interviews were coded and the participants were given pseudonyms in all reports from this study to ensure anonymity and confidentiality.

**Rigour.** (1) The author first learned the knowledge of qualitative study through the graduate classes, and then learned the books and literature on qualitative study to have a more in-depth understanding of the implementation of research. (2) To establish a good interview environment, the first author participated in the process of patient care in the hospital to understand the patient's condition, the personality of the patients and their caregivers, family background, and so on. (3) During the interview, the first author maintained an objective and neutral attitude and avoided suggestive questions. Interview materials were confirmed by the interviewees to ensure that the recorded results were consistent with the their statement. In the process of data analysis, the original statements of the interviewees did not mix with personal

understanding and opinions. (4) The first interviewer conducted all the interviews primarily, ensuring authenticity and data saturation through prolonged engagement.

**Data analysis.** No instrument or software that requires copyright was used in this study. Convert the recording into an electronic file within 24 hours after each interview. A conventional content analysis [23] was performed in this study, including the following: ①The interviews were read repeatedly to obtain a sense of the content in its entirety. ②Meaning units (text segments that convey interesting information about the research question) were derived from the text, condensed, and labeled with a code capturing the key concept of the text. ③The codes were abstracted and sorted into subcategories and categories based on how the different codes were related. ④The interviews were read again to check for the adequacy of the categories and subcategories regarding the content of the interviews. ⑤Citations for each subcategory were identified from the data. The summaries were descriptive regarding the content of the discussions. Two researchers analyzed and compared the data simultaneously to improve the quality of data analysis in this study.

## Findings

The final sample consisted of 6 PSD patients(labelled 'P' in quotes), who ranged from 39 to 82 years, and 11 family caregivers(labelled 'C' in quotes), ranging in age between 24 and 64 years, in the face-to-face interview. A total of 16 participants took part in the first month's telephone follow-up interview(one patient was not involved due to death by aggravation of illness), 14 participants in the third month(3 patients were not involved due to death by aggravation of illness) and 12 participants in the sixth month(3 patients were not involved due to death by aggravation of illness and 2 had no new transition experience, and the interview content was the same as that in the third month). General information of interviewees see Table 1.

The conventional content analysis revealed that the themes derived from the patients and family caregivers were broadly consistent, which reflects the integrity of patients and family caregivers. Three themes and ten subthemes were identified in the analysis(Fig 2). These themes and subthemes were illustrated by the text and quotations below. S2 File presents the identified themes and the corresponding interviews.

## Transition from onset to admission

**Stroke occurred in an unexpected and abrupt way.** Stroke occurs suddenly and can occur at any time, and there is no way to determine a period with a higher frequency of occurrence. Patients and their family caregivers said that the onset of stroke made them feel unprepared and unexpected.

> P4: "*After I woke up to wash my face, I saw that my face was different. I wanted to speak, but I couldn't say anything, and my hands couldn't lift.*"

> F7: "*It happened so suddenly. We were having dinner, and we saw that my husband couldn't walk well and couldn't speak clearly.*"

**Inadequate dysphagia related knowledge and information.** Most participants paid little attention to swallowing function, especially those who were older because they thought that it was the decline in swallowing function caused by old age. They didn't pay enough attention to the influence of PSD.

**Table 1. Participant general characteristics.**

| ID | Age | Gender | Marital status | Transitions experienced | Relationship to patient | Number of participants in interviews | | |
|----|-----|--------|----------------|-------------------------|-------------------------|-------------------------|---------------------|---------------------|
| | | | | | | The first month(16) | The third month(14) | The sixth month(12) |
| P1 | 56 | Male | Married | R, H | Himself | Y | Y | Y |
| P2 | 59 | Male | Married | R, H | Himself | Y | Y | Y |
| P3 | 52 | Male | Married | R, H | Himself | Y | Y | Y |
| P4 | 55 | Male | Married | R, H | Himself | Y | Y | Y |
| P5 | 62 | Male | Married | H | Himself | Y | Y | Y |
| P6 | 49 | Female | Married | H | Herself | Y | Y | N |
| F1 | 51 | Female | Married | R, H | Daughter | Y | Y | Y |
| F2 | 42 | Male | Married | H | Husband | Y | Y | Y |
| F3 | 30 | Male | Married | R | Son-in-law | Y | D | D |
| F4 | 63 | Female | Married | H | Wife | Y | Y | N |
| F5 | 56 | Male | Married | R | Son | Y | Y | Y |
| F6 | 24 | Female | Single | R, H | Granddaughter | Y | Y | Y |
| F7 | 52 | Female | Married | R | Wife | Y | Y | Y |
| F8 | 35 | Male | Married | R | Son | Y | D | D |
| F9 | 44 | Female | Married | R | Wife | D | D | D |
| F10 | 32 | Male | Married | R, H | Nephew | Y | Y | Y |
| F11 | 64 | Male | Married | R | Cousin | Y | Y | Y |

Abbreviations: R = rehabilitation institutions, H = home, Y = participation, D = death, N = non-participation.

F4: "*He was not so serious on the day of onset. We thought he just choked, but now he couldn't eat anything.*"

P2: "*At first I thought I just had no appetite, and it was common to choke when you were old. However, look at me now, I couldn't even drink water.*"

Family caregivers were just recovered from the initial shock of their family members' stroke and took care of survivors without any professional training during this time. They pointed

**Fig 2. Themes and subthemes derived from interviews with patients with post stroke dysphagia and family caregivers.**

out that there were still questions about the choice and preparation of food and what needed to be done when aspiration occurred during oral feeding.

> F2: "*The doctor said that she has dysphagia, but we would rather eat slowly than inserting a gastric tube. I think the most difficult part is what kind of food we can choose, and we are afraid that his nutrition can't keep up.*"

> F10: "*It is scary when my uncle drinks water because he sometimes chokes badly. His face will turn purple and we don't know what to do. So we hope that medical employees can teach us how to deal with choking before leaving the hospital.*"

Most patients chose to keep a gastric tube after being assessed as dysphagia, and the participants expressed their desire to learn more about the knowledge of tube feeding.

> F6: "*Now he can't eat by himself, and he can only get food from the stomach tube. If we have to keep the tube when we leave the hospital, how can we take care of it at home? If the tube is pulled out, can we plug it back in ourselves?*"

> F11: "*Is it possible to replace this tape for fixing the stomach tube with something else? He has to put this tape on every day, which makes his face red all over. I'm worried that it will break his skin. Is there any other way?*"

Moreover, participants also want to learn the information regarding home preparation, prevention of complications, and making follow-up appointments.

**Cooperating with swallowing rehabilitation training actively.** In general, patients begin swallowing rehabilitation training during hospitalization. They looked for information regarding their own situation and what they must do to improve, actively participating in the swallowing rehabilitation training, in an attempt to regain their abilities and reduce the burden on their families.

> P4: "*I want to recover as soon as possible, especially my swallowing function, otherwise you can't enjoy a lot of things. The food is injected directly into the stomach, and even the best things have no feeling.*"

> P6: "*My family has worked hard to take care of me. I will cooperate with all treatment and training I need. I want to recover soon.*"

**Impact of dysphagia.** The patient's eating pleasure was reduced due to dysphagia after onset, and some hobbies were forced to be put down. For caregivers, the role of caregivers leaded to reduced communication with the outside world and less time for recreation, socialization and self-realization.

> P5: "*You can't enjoy a lot of things anymore. If you just pour it directly into your stomach, you won't feel anything even if it's the best thing.*"

> P4: "*I used to speak in a clear and melodious voice, but now it's like I were thirty years older. I loved to sing, look at this voice now, I can't sing anymore.*"

## Transition from discharge to other rehabilitation institutions

In our study, a total of 13 patients and family caregivers have experienced the transition to other rehabilitation institutions and drew three sub-themes.

**Incomplete handover of medical information and lack of information sharing platform.** Due to the imperfect information platform of hospitals at different levels, the sharing of medical information has not become a reality, and the patient's illness information was interrupted under the condition of incomplete handover of medical information. It may increase the frequency of swallowing function evaluation of patients, thus increasing the risk of aspiration. In addition, different swallowing rehabilitation training methods may also lead to different training effects.

> F7: "*We already had a swallowing assessment done at the hospital before we were transferred, it failed. And after we were transferred, the nurses here did it again, and it still failed. so why do we have to do it all over again? I was afraid that he(the patient) would choke.*"

> F6: "*I felt that swallowing rehabilitation training was quite good in your hospital before, but after the transfer, the doctor here gave she(patient) a new training method, so I was a little worried. What if this rehabilitation effect is not as good as before?*"

**Gap in swallowing rehabilitation nursing services between hospitals.** For patients and families transitioning to other healthcare facilities for rehabilitation, they may be concerned about the poorer professionalism of the healthcare staff at the lower level hospitals and the effectiveness of swallowing rehabilitation training.

> F8: "*We just hope that my dad can be cured in your hospital, and we shouldn't be transferred to other hospitals. We heard that your hospital is the best hospital for cerebrovascular diseases. Will the risk be higher if my dad is transferred out?*"

> P1: "*I can't see my attending doctor every day here. The doctors in your hospital are very detailed, that is, every doctor knows the condition very well, which may not be the case here.*"

**Differences in ward environments.** Some patients who were transferred to other hospitals for rehabilitation considered that rehabilitation hospitals were relatively less crowded, more spacious and quieter, which was more conducive to rehabilitation. They also hoped that higher-level hospitals can improve the hospitalization environment and enhance the comfort of hospitalization.

> P4: "*It's a good environment here, not as crowded as the hospital before. I can do some training back and forth in the corridor.*"

> P2: "*The wards in lower-level hospitals feel more spacious and quieter, which is suitable for recuperation.*"

## Transition from discharge to home

There were 11 patients have experienced the transition from discharge to home and three sub-themes were analyzed.

**Continuous nursing services for dysphagia.** At the sixth month follow-up, nine patients had recovered from dysphagia, and the participants who had returned home were more interested in learning about self-management practices related to PSD and methods of preventing recurrence in various convenient ways.

> P1: "*When she(the patient) was discharged, the nurse told us a lot of precautions, but what she(the nurse) said was too much for us to remember, and when we got home. I think the WeChat follow-up visit is very good, but sometimes there was no reply when asked.*"

*F10*: "*Most importantly, we're worried about his swallowing function. It would be nice if you could tell us when it will recover.*"

Patients and their family caregivers had not learned professional care skills, and they were expected to obtain detailed guidance during the nursing process when they were discharged and went home, especially elderly individuals.

*P3*: "*After coming back from the hospital, my wife is the one who takes care of me. There are only us two at home, and we haven't learned professional knowledge, such as management of aspiration. Sometimes we don't know whether we have done it right or not. I hope you can give more guidance in this respect.*"

*P5*: "*For example, you said I needed a high-protein diet, but we didn't know what high-protein foods were. You should arrange nurses to teach us, like some precautions and so on.*"

**Lack of specialized swallowing rehabilitation training institutions.**   Some participants said that they wanted to go to tertiary hospitals for rehabilitation training again after leaving the hospital, but it was difficult to make appointments. They generally gave priority to tertiary hospitals because they were worried about the effect of rehabilitation training in secondary and community hospitals.

*F5*: "*You know, we want to go to your hospital for rehabilitation training, but we haven't been able to make an appointment, and we don't want to go to other hospitals. What should we do?*"

*F11*: "*At that time, his swallowing function had not fully recovered, and we were still worried after he was discharged from the hospital. It was more troublesome to come back for a follow-up visit and do rehabilitation training, for example, we waited for several days to reserve a bed.*"

**Psychological changes in the long-term rehabilitation process.**   Five patients failed to recover their swallowing function, indicating that there was little hope for their recovery during the long rehabilitation process.

*P2*: "*I've given up now, it's been so long, and it can't be cured. Nothing has changed. Oh well, so be it…*"

*F1*: "*We take turns taking care of her at home. All of us have to go to work, and we all take care of her during our rest time, but she hasn't been well for so long. We are truly tired (sobbing).*"

## Discussion

Based on Meleis's transition theory, this longitudinal study provided a vivid evidence for the experiences of patients with post stroke dysphagia in three transition stages. Also, the experiences of the patients and family caregivers were consistent, which reflects the integrity of patients and caregivers. This suggests that patients and family caregivers should be regarded as an entirety in the nursing process. In the framework of Meleis's transition theory, PSD patients and family caregivers have experienced different types of transitions, including health and illness(change of health status), developmental(change of role), situational(change of

environment) at the stage of admission and discharge to home, and situational(change of hospital environment) and organizational(change of hospital culture and ward culture) at the stage of transfer to other medical institutions. Various types were interrelated, and some appeared at the same time. From the interview results, the factors facilitating the smooth transition of patients were the active cooperation of patients' rehabilitation training, comfortable hospitalization environment and long-term personalized continuous nursing service, while the factors inhibiting the transition included insufficient understanding of post stroke dysphagia, inadequate discharge preparation, lack of perfect information sharing among medical institutions, and shortage of medical resources. The nursing therapy patients received included swallowing rehabilitation training during hospitalization, health education before discharge, and telephone or WeChat follow-up after discharge.

It was found in this study that PSD patients and family caregivers lack proper understanding of PSD, and when patients can't eat orally, they may feel anxious and helpless, worrying about the prognosis of swallowing function, which was similar to the results of Allen et al [24]. We also found that some patients and their families perceived that other challenges such as mobility difficulties, impacted their lives more than dysphagia and managing dysphagia was a lower priority, which was similar to the conclusion drawn by Howells et al [25]. The reason may be that the influence of dysphagia is not as great as that of other disorders. And with increasing age, the sensory function related to swallowing gradually declines and the pharyngeal reflex weakens [26]. Patients and their families may think that dysphagia is a natural phenomenon caused by old age. In this case, a comprehensive health education should be given to patients and family caregivers. In addition to the prognostic information of PSD, they should also learn that swallowing function is not only closely related to patients' eating and nutritional status, but also affects patients' social interaction and self-image [27]. A proper understanding of PSD should be established.

Statistics [28] have shown that 53.1% of stroke patients have functional dependence at discharge. Lee et al. [29] found that 17.5% of stroke patients still have dysphagia after 6 months. In our study, 5 patients still had dysphagia after 6 months, which indicated that dysphagia lasted for a long time. Affected by policies such as shortening the average length of hospital stay and early discharge, more than 90% of PSD survivors choose to go home for rehabilitation nursing after the acute phase in the hospital [30]. This means that patients' rehabilitation care mostly depends on the community and family, but the community rehabilitation resources and management system in China are still not perfect [31]. As a result, patients' demand for discharge plan service and continuing nursing is more prominent. Multidisciplinary teams should cooperate to fully utilize nursing resources at all levels and provide patients with a continuous service model of diagnosis, treatment, rehabilitation and long-term care. Patients and family caregivers have dyadic wholeness [32], and family caregivers should be encouraged to actively participate in the nursing process of patients. Nursing skills such as oral feeding, tube feeding, prevention and treatment of aspiration need to be taught to patients and family caregivers. Moreover, patients' demands for rehabilitation care are dynamically changing with individual differences and disease progress [33], so continuing nursing and follow-up should be based on patients' real needs instead of preset protocol.

The transitional experience of PSD patients and their caregivers showed that the transfer of referral information between medical institutions at all levels needs to be improved. There was a lack of information sharing platforms between different levels of hoapitals and led to the fragmentation of patients' health information [34], which may increase the frequency of swallowing function evaluation and the risk of aspiration. Discontinuous information transmission would also lead to a decline in patient satisfaction [35] and influencing the effectiveness of swallowing rehabilitation training. This may require the government to build an information

sharing platform to promote the sharing and exchange of medical and health information to effectively avoid repeated examination and reduce the risk of aspiration, ultimately improve the quality of nursing services in lower-level hospitals.

The patients' swallowing function had not recovered for a long time, and in addition to the nutritional status of the patient, the confidence of the patient and caregivers was also affected. The longer the rehabilitation was, the more likely negative emotions would be generated. This was because the patient's condition still did not improve, and it was unknown whether the patient could recover, which made the patients and family caregivers feel agitated or even desperate. This fingding was similar to the results of Oliva-Moreno et al [36]. Hence, similar successful cases should also be introduced to patients and their family caregivers who have been recovering for a long time to help them maintain a positive attitude and increase their confidence in rehabilitation. If possible, patients with similar experiences can be invited to share their ways to success.

The study has some limitations. First, the study was severely limited by the low number of PSD patients and family caregivers enrolled, which may prevent from drawing some important conclusions. Second, the study was conducted at a single center, and some views came from the family caregiver's perspective, but these family caregivers were closely associated with the patient. Therefore, the results of the interviews were also reliable. Third, the interviews after discharge were conducted via telephone, so the feelings of the patients and their caregivers while describing their experiences could not be understood by facial expressions or gestures.

## Conclusions

Based on Meleis's transition theory, PSD patients and their family caregivers have experienced multiple, simultaneous, and interrelated transitions, which may be complex and multifaceted. There were also various facilitating and inhibiting factors. To help survivors with post stroke dysphagia and family caregivers facilitate their transition trajectory, collaboration with health care professionals, accessible rehabilitation services and follow-up support after discharge, and available community and social support should be integrated into transitional care. It is important to be aware of the patients' changing experiences and nursing needs and to support the necessary adaptations at different transitions.

## Supporting information

**S1 Checklist. Consolidated criteria for reporting qualitative research, the COREQ guidelines were followed when reporting the study.**
(DOCX)

**S1 File. Topic guide informed by the main concepts of meleis's transition theory.**
(DOC)

**S2 File. Themes and the corresponding interviews.**
(DOC)

## Acknowledgments

The authors would like to thank all of the patients and their families for their participation in the study and the staff associated with the study's data collection and management.

## Author Contributions

**Conceptualization:** Jian CHEN, Xiaomei Zhang.

**Data curation:** Yuan WANG, Lin LIAO, Mingyu YAN.

**Formal analysis:** Jian CHEN, Jianhui CHEN.

**Funding acquisition:** Xiaomei Zhang.

**Investigation:** Jian CHEN, Jianhui CHEN.

**Methodology:** Jian CHEN, Jianhui CHEN.

**Project administration:** Yansi LUO.

**Resources:** Yansi LUO.

**Supervision:** Xiaomei Zhang.

**Writing – original draft:** Jian CHEN, Jianhui CHEN.

**Writing – review & editing:** Jian CHEN, Jianhui CHEN, Yanli CUI, Yansi LUO, Xiaomei Zhang.

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
