## [Decision Letter · Decision Letter 0]

20 Feb 2024

PONE-D-23-41035Transition experiences of patients with post-stroke dysphagia: a longitudinal, qualitative studyPLOS ONE

Dear Dr. Zhang,

Thank you for submitting your manuscript to PLOS ONE. After careful consideration, we feel that it has merit but does not fully meet PLOS ONE’s publication criteria as it currently stands. Therefore, we invite you to submit a revised version of the manuscript that addresses the points raised during the review process.

We look forward to receiving your revised manuscript.

Kind regards,

Antonino Maniaci

Academic Editor

PLOS ONE

Journal Requirements:

" ext-link-type="uri" xlink:type="simple">https://journals.plos.org/plosone/s/file?id=ba62/PLOSOne_formatting_sample_title_authors_affiliations.pdf"

This work was supported by grants from the Natural Science Foundation of Guangdong Province (grant number 2022A1515012184).

3. In the online submission form, you indicated that Data cannot be shared publicly because of confidentiality of patient information. The data used and/or analyzed during the current study are available from the corresponding author on reasonable request.

Additional Editor Comments:

Please perform all the revisions suggested.

Reviewers' comments:

Reviewer's Responses to Questions

**Comments to the Author**

1. Is the manuscript technically sound, and do the data support the conclusions?

Reviewer #1: Partly

Reviewer #2: Yes

2. Has the statistical analysis been performed appropriately and rigorously? 

Reviewer #1: N/A

Reviewer #2: No

3. Have the authors made all data underlying the findings in their manuscript fully available?

Reviewer #1: No

Reviewer #2: No

4. Is the manuscript presented in an intelligible fashion and written in standard English?

Reviewer #1: No

Reviewer #2: Yes

5. Review Comments to the Author

Reviewer #1: I read with interest the manuscript by CHEN et al. Here there are some comments to improve the manuscript:

- The manuscript must be edited for English.

- Line 83. What do you mean by descriptive qualitative study? Please specify whether the study is retrospective or prospective. In fact, the study seems more like a case series.

- It is not clear to me whether the caregivers were included for all participants, or just for those who could not communicate.

- Line 105. How was this training performed?

- In the limitation section, authors should report that the study is severely limited by the low number of patients and caregivers enrolled, which prevents from drawing any conclusion. Also, the lack of control group, the lack of statistical analyses are also limitations to be presented.

- Conclusions must be limited to the very small cohort of patients included, and could not be generalized

Reviewer #2: I have reviewed the manuscript and found some areas for improvement:

The manuscript presents intriguing findings that could pave the way for further research in the field. It is well-written, with a concise and informative abstract.

While the topic is well-justified, it would be beneficial to understand how local healthcare settings manage post-stroke depression (PSD) patients. Additionally, incorporating a conceptual framework derived from transitional theory could enhance the theoretical underpinning of the study.

Considering the involvement of caregivers in the study, a slight modification to the title to include the keyword "caregivers' perception" may be warranted. Consequently, adjustments to the interview guide to accommodate caregivers' perspectives are necessary. To ensure clarity, it's important to explain in detail the longitudinal qualitative approach used in the study and refrain from numbering descriptions to avoid confusion with citation numbers. Structuring the in-depth interviews into proper paragraphs would enhance the rigor and trustworthiness of the data.

The inclusion of follow-up elements, as indicated in Table 3, suggests a longitudinal design approach. However, clarification is needed regarding the inclusion of participants who passed away during the study. Additionally, explanations are required regarding the varying number of interviews per participant, considering that qualitative interviews typically continue until data saturation is reached.

Table 4 suggests that all patients may need to be referred to other rehab centers for further evaluation, but this needs further elaboration. Regarding the quotes and evidence presented, it's essential to ensure alignment with the study's focus on stroke and dysphagia, rather than general stroke issues. Contradictory statements within subthemes should be addressed, and inaccuracies in explaining certain themes need correction.

Theme 2 should focus on the procedure for managing post-stroke patients, including whether referral to rehab centers or lower-level hospitals before discharge is necessary. It's important to balance perspectives between patients and caregivers, with a priority given to the patients' viewpoint, as per the study's title. Any inaccuracies in subtheme explanations should be rectified to maintain coherence and accuracy throughout the discussion.

6. PLOS authors have the option to publish the peer review history of their article (what does this mean?). If published, this will include your full peer review and any attached files.

Reviewer #1: No

Reviewer #2: No

---

## [Author Response · Author response to Decision Letter 0]

30 Apr 2024

Dear Editors and Reviewers:

We would like to thank both reviewers and the editor for their positive review of our article entitled “Transition experiences of patients with post stroke dysphagia and family caregivers: a longitudinal, qualitative study” (ID: PONE-D-23-41035). Those comments are all valuable and very helpful for revising and improving our paper, as well as the important guiding significance to our article. We have studied the comments carefully and have made corrections that we hope will be met with approval. We have listed comments below and our responses. These have been highlighted in the revised manuscript. The main corrections in the paper and the responds to the comments are as flowing:

Journal Requirements:

1.Please ensure that your manuscript meets PLOS ONE’s style requirements, including those for file naming.

Reply: Thank you for making this valuable suggestion. We have revised the format of the manuscript according to the PLOS ONE’s style requirements.

2.Please provide an amended statement that declares all the funding or sources of support (whether external or internal to your organization) received during this study. Please also include the statement “There was no additional external funding received for this study.” in your updated Funding Statement. 

Reply: Thank you for this. This study was supported by only one funding and we have amended the statement and declared in our cover letter this study was supported by grants from the Natural Science Foundation of Guangdong Province and the funders had no role in study design, data collection and analysis, decision to publish, or preparation of the manuscript. 

Funding statement:

This work was supported by grant from the Natural Science Foundation of Guangdong Province (grant number 2022A1515012184). The funders had no role in study design, data collection and analysis, decision to publish, or preparation of the manuscript.

3.All PLOS journals now require all data underlying the findings described in their manuscript to be freely available to other researchers, either a. In a public repository, b. Within the manuscript itself, or c. Uploaded as supplementary information.

Reply: The data underlying the results of this study are available upon request due to ethical and legal restrictions under the medical ethics committee of NanFang hospital. Interested researchers may contact the authors at 1329319886@qq.com or manager of medical ethics committee of NanFang hospital at nfyyec@163.com. We do not have the authority to provide personal information such as the names of the participants because of the principle of confidentiality, but we have provided relevant interview transcripts from the participants needed for the analysis of themes in S3 file. 

Review Comments to the Author:

Reviewer #1: I read with interest the manuscript by CHEN et al. Here there are some comments to improve the manuscript:

1.The manuscript must be edited for English.

Reply: We feel sorry for the inconvenience brought to you. We have modified the expression throughout the article. 

2.Line 83. What do you mean by descriptive qualitative study? Please specify whether the study is retrospective or prospective. In fact, the study seems more like a case series.

Reply: We feel sorry for the inconvenience brought to you. Descriptive qualitative study is a research approach based in the philosophical tenets of naturalistic inquiry. It’s features are recognition of varied shared experiences and the interactive-inseparable nature of human interaction and is analyzed using a content analysis method. Related introduction of descriptive qualitative study has been added on page 6, lines 112-118. 

In this study, we conducted face-to-face interviews with participants, and then the transition experiences after discharge were collected prospectively by telephone follow-up interviews. So we believe the study is prospective. We have specified this on page 7, lines 147-157.

3.It is not clear to me whether the caregivers were included for all participants, or just for those who could not communicate.

Reply: We feel sorry for the inconvenience brought to you. This is one of the studies of a project titled Construction of hospital-community integrated continuous rehabilitation nursing program for patients with post stroke dysphagia, and the main object was patient with post stroke dysphagia. Stroke patients may have concomitant dysarthria, speech impairment, and impaired consciousness, and the transition experiences of this population should also be explored. So we will select family caregivers according to the specific situation of PSD patients. Family caregiver is considered for inclusion if patient who meet the inclusion and exclusion criteria is unable to communicate properly. We have explained this on page 6-7, lines 127-131.

4.Line 105. How was this training performed?

Reply: We feel sorry for the inconvenience brought to you. The training mentioned here refers to the fact that the researcher has systematically learned about qualitative study method through a postgraduate classes and has reviewed books and literature related to qualitative study to understand the implementation of the study in order to facilitate the smooth implementation of the study. We have added this on page 9, lines 183-186. 

5.In the limitation section, authors should report that the study is severely limited by the low number of patients and caregivers enrolled, which prevents from drawing any conclusion. Also, the lack of control group, the lack of statistical analyses are also limitations to be presented.

Reply: Thank you for making this valuable suggestion. We have added to the limitations section the failure to draw full conclusions due to the low number of patients and caregivers included in the study on page 24, lines 492-494. 

The qualitative study refers to an activity in which the researcher himself is used as a research tool, and various data collection methods are used to explore the research phenomenon as a whole in the natural situation, with text narration as the material and induction as the demonstration step, and useful information is obtained through interaction with the research object. The qualitative study was designed without a control group and statistical analyses, so we did not set it up when implementing the research. Accordingly, the lack of control group and the lack of statistical analyses have not been added to the limitations. 

6.Conclusions must be limited to the very small cohort of patients included, and could not be generalized.

Reply: Thank you for making this valuable suggestion. We have revised our conclusions based on modified Findings and Discussion on page 24, line 505-511. 

Reviewer #2: I have reviewed the manuscript and found some areas for improvement:

The manuscript presents intriguing findings that could pave the way for further research in the field. It is well-written, with a concise and informative abstract.

1.While the topic is well-justified, it would be beneficial to understand how local healthcare settings manage post-stroke depression(PSD) patients. Additionally, incorporating a conceptual framework derived from transitional theory could enhance the theoretical underpinning of the study.

Reply: Thank you for making this valuable suggestion. After reading the relevant literature, we found that the management of post stroke dysphagia mainly includes risk factors for dysphagia, early identification, management of aspiration, and telephone follow-up in most local healthcare settings. We have added this on page 4, lines 72-74. The conceptual framework derived from transitional theory has been presented in Figure 1.

2.Considering the involvement of caregivers in the study, a slight modification to the title to include the keyword “caregivers” perception may be warranted. Consequently, adjustments to the interview guide to accommodate caregivers perspectives are necessary.

Reply: Thank you for making this valuable suggestion. We completely agree with the reviewer and the title have been modified to “Transition experiences of patients with post stroke dysphagia and family caregivers: a longitudinal, qualitative study”. The interview guide was adjusted accordingly, as detailed in S2 file.

3.To ensure clarity, it’s important to explain in detail the longitudinal qualitative approach used in the study and refrain from numbering descriptions to avoid confusion with citation numbers. Structuring the in-depth interviews into proper paragraphs would enhance the rigor and trustworthiness of the data.

Reply: Thank you for making this valuable suggestion. Descriptive qualitative study is a research approach based in the philosophical tenets of naturalistic inquiry. It’s features are recognition of varied shared experiences and the interactive-inseparable nature of human interaction and is analyzed using a content analysis method. Related introduction of descriptive qualitative study has been added on page 6, lines 112-118. 

We apologize for any inconvenience caused by the formatting and numbering of paragraphs throughout the article. All numbering in the article has been adjusted to avoid confusion with citation numbers. Also, the paragraphs of the in-depth interviews have been adjusted.

4.The inclusion of follow-up elements, as indicated in Table 3, suggests a longitudinal design approach. However, clarification is needed regarding the inclusion of participants who passed away during the study. Additionally, explanations are required regarding the varying number of interviews per participant, considering that qualitative interviews typically continue until data saturation is reached.

Reply: Thank you for making this valuable suggestion. When a telephone follow-up reveals that a patient has died, we did not interview the family caregiverin order not to increase their pain. We have added the number of participants in each follow-up interview, the number of participants who failed to participate in the follow-up interview and the reasons for their loss on page 10-11, line 214-223. 

5.Table 4 suggests that all patients may need to be referred to other rehab centers for further evaluation, but this needs further elaboration. Regarding the quotes and evidence presented, it’s essential to ensure alignment with the study’s focus on stroke and dysphagia, rather than general stroke issues. Contradictory statements within subthemes should be addressed, and inaccuracies in explaining certain themes need correction.

Reply: Thank you for making this valuable suggestion. In our study, a total of 13 patients have experienced the transition to other rehabilitation institutions, 11 patients have experienced the transition from discharge to home. We have added the data in Table 1 and on page 15, line 308-310, page 17, line 353-354. 

We have modified the content of the themes in the revised manuscript. While some of these themes may be common to stroke patients and family caregivers in the transition process rather than specific to dysphagia, we believe that these were also part of their transition experiences. Rehabilitation of swallowing function can also be facilitated if improvements are made to these experiences such as the improvement of nursing quality and ward environment.

6.Theme 2 should focus on the procedure for managing post-stroke patients, including whether referral to rehab centers or lower-level hospitals before discharge is necessary. It’s important to balance perspectives between patients and caregivers, with a priority given to the patients’ viewpoint, as per the study’s title. Any inaccuracies in subtheme explanations should be rectified to maintain coherence and accuracy throughout the discussion.

Reply: Thank you for making this valuable suggestion. We have modified Theme 2 to focus more on post stroke dysphagia. See page 15-17, line 307-350. 

As for balancing perspectives between patients and caregivers, we have modified the title to “Transition experiences of patients with post stroke dysphagia and family caregivers: a longitudinal, qualitative study”. So it may be a bit inappropriate to give priority to the patients’ viewpoint. We apologize for not taking this comment. 

And we have modified the content of the themes in the revised manuscript to ensure a better fit with the article topic.

---

## [Decision Letter · Decision Letter 1]

10 May 2024

Transition experiences of patients with post stroke dysphagia and family caregivers: a longitudinal, qualitative study

PONE-D-23-41035R1

Dear Dr. Zhang,

We’re pleased to inform you that your manuscript has been judged scientifically suitable for publication and will be formally accepted for publication once it meets all outstanding technical requirements.

Kind regards,

Keisuke Suzuki, MD, PhD

Academic Editor

PLOS ONE

Additional Editor Comments (optional):

Reviewers' comments:

Reviewer's Responses to Questions

**Comments to the Author**

1. If the authors have adequately addressed your comments raised in a previous round of review and you feel that this manuscript is now acceptable for publication, you may indicate that here to bypass the “Comments to the Author” section, enter your conflict of interest statement in the “Confidential to Editor” section, and submit your "Accept" recommendation.

Reviewer #1: (No Response)

Reviewer #2: All comments have been addressed

2. Is the manuscript technically sound, and do the data support the conclusions?

Reviewer #1: (No Response)

Reviewer #2: Yes

3. Has the statistical analysis been performed appropriately and rigorously? 

Reviewer #1: (No Response)

Reviewer #2: Yes

4. Have the authors made all data underlying the findings in their manuscript fully available?

Reviewer #1: (No Response)

Reviewer #2: No

5. Is the manuscript presented in an intelligible fashion and written in standard English?

Reviewer #1: (No Response)

Reviewer #2: Yes

6. Review Comments to the Author

Reviewer #1: The authors successfully addressed all the comments provided. I believe the manuscript is now suitable for publication.

Reviewer #2: I have reviewed the revised version of the manuscript and found that all previously raised concerns have been thoroughly addressed by the authors. The current version of the manuscript exhibits significant improvement, offering enhanced clarity and readability. It merits recommendation for publication.

7. PLOS authors have the option to publish the peer review history of their article (what does this mean?). If published, this will include your full peer review and any attached files.

Reviewer #1: No

Reviewer #2: No

---

## [Editor Report · Acceptance letter]

24 May 2024

PONE-D-23-41035R1 

PLOS ONE

Dear Dr. Zhang, 

I'm pleased to inform you that your manuscript has been deemed suitable for publication in PLOS ONE. Congratulations! Your manuscript is now being handed over to our production team.

Kind regards, 

on behalf of

Dr. Keisuke Suzuki 

Academic Editor

PLOS ONE